# SGLT2 Inhibitors in Acute Heart Failure: A Meta-Analysis of Randomized Controlled Trials

**DOI:** 10.3390/healthcare10122356

**Published:** 2022-11-23

**Authors:** Noor Ul Amin, Faiza Sabir, Talal Amin, Zouina Sarfraz, Azza Sarfraz, Karla Robles-Velasco, Ivan Cherrez-Ojeda

**Affiliations:** 1Department of Acute Medicine, King’s Mill Hospital, Sutton-in-Ashfield NG17 4JL, UK; 2Department of Research, King Edward Medical University, Lahore 54000, Pakistan; 3Department of Research, Nishtar Medical College, Multan 60000, Pakistan; 4Department of Research and Publications, Fatima Jinnah Medical University, Lahore 54000, Pakistan; 5Department of Pediatrics and Child Health, The Aga Khan University, Karachi 74800, Pakistan; 6Department of Allergy, Immunology & Pulmonary Medicine, Universidad Espíritu Santo, Samborondón 092301, Ecuador

**Keywords:** acute heart failure, SGLT2 inhibitors, cardioprotection, cardiovascular mortality, heart failure events

## Abstract

Acute heart failure (AHF) is a major public health concern, affecting 26 million worldwide. Sodium-glucose cotransporter 2 (SGLT2) inhibitors are a class of glucose-lowering drugs, comprising canagliflozin, dapagliflozin, and empagliflozin that are being explored for AHF. We aim to meta-analyze the effectiveness of SGLT2 inhibitors compared to placebo for primary outcomes including all-cause and cardiovascular mortality, heart failure events, symptomatic improvement, and readmissions. Our secondary outcome is the risk of serious adverse events. This meta-analysis has been designed in accordance with the PRISMA Statement 2020. A systematic search across PubMed, Scopus, and Cochrane Library was conducted through August 13, 2022. The following keywords were utilized: sglt2, sodium-glucose transporter 2 inhibitors, sglt2 inhibitors, decompensated heart failure, de-novo heart failure, and/or acute heart failure. Only randomized controlled trials (RCTs) with adult patients (>18 years), hospitalized with de-novo AHF, acutely decompensated chronic heart failure with reduced, borderline, or preserved ejection, and receiving SGLT2 inhibitors were included. A quantitative analytical methodology was applied where the standardized mean difference (SMD) applying 95% confidence intervals (CI) for continuous outcomes and risk ratio (RR) with 95% CI was yielded. All tests were carried out on Review Manager 5.4 (Cochrane). In total, three RCTs were included pooling in a total of 1831 patients where 49.9% received SGLT2 inhibitors. The mean age was 72.9 years in the interventional group compared to 70.6 years in the placebo. Only 33.7% of the sample was female. The follow-up spanned 2–9 months. Heart failure events were reduced by 62% in the interventional group (RR = 0.66, *p* < 0.0001). readmissions had a reduced risk of 24% with SGLT2 inhibitors (RR = 0.76, *p* = 0.03). We assessed the efficacy and safety of SGLT2 inhibitors in preventing complications post-AHF. The odds of all-cause mortality, cardiovascular mortality, heart failure events, and re-admissions rates were substantially reduced within the first 1–9 months of hospitalization.

## 1. Introduction

Acute heart failure (AHF) is a major public health problem, affecting nearly 6 million people in the United States and 26 million people globally [1]. AHF is defined as a rapid onset or worsening of signs and symptoms of heart failure (HF) that require urgent medical care [2]. With over 1 million annual hospitalizations in the US, AHF is the leading cause of admissions in the elderly [3]. Approximately 1 in 5 patients with AHF are readmitted within 30 days of being discharged and over 3 in 5 patients are readmitted during the first year [4]. The 1-year mortality rate is 10–30% with the highest risk of mortality within 30 days of the index hospitalization [5,6,7]. Despite therapeutic advances in chronic heart failure (CHF) [8], AHF has a dismal prognosis since no therapy is proven to have long-term mortality benefits [9]. Loop diuretics are the mainstay of therapy for AHF but have a negative impact on renal function as well as potential neurohormonal imbalance [10,11,12].

Sodium-glucose cotransporter 2 (SGLT2) inhibitors are a new class of glucose-lowering drugs, including canagliflozin, dapagliflozin, and empagliflozin, that block the SGLT2 protein located in the proximal convoluted tubule of the nephron for adults with type 2 diabetes [13]. The SGLT2 protein is responsible for the resorption of nearly 90% of filtered glucose [14,15,16,17]. In 2015, among patients with cardiovascular disease (CVD), the Empagliflozin Outcome Event Trial in Type 2 Diabetes Mellitus Patients–Removing Excess Glucose (EMPA-REG OUTCOME) trial indicated a significant reduction in composite risk of cardiovascular death, myocardial infarction, or stroke by 14%; the risk of all-cause mortality was reduced by 32% over a mean duration of 3.1 years [18,19]. While such promising evidence was encouraging, there were concerns about several adverse events that include acute kidney injury (AKI), diabetic ketoacidoses (DKA), and urinary tract infections (UTIs) based on initial reports [18,19]. Since then, many randomized controlled trials (RCTs) evaluated SGLT2 as a potential adjunctive pharmacotherapy for HF patients, which has shown significant benefits. SGLT2 inhibitors, empagliflozin, and dapagliflozin demonstrated efficacy for patients with CHF who had a reduced left ventricular ejection fraction (LVEF) [20,21]. More recently, the U.S. Food and Drug Administration (FDA) approved empagliflozin in February 2022, for patients with HF regardless of LVEF to reduce the risk of cardiovascular death and hospitalization [22].

Whether SGLT2 inhibitors provide clinical benefits in patients with AHF is being explored. Despite emerging trials, there are limitations in systematically assessing the efficacy and safety of SGLT2 inhibitors among patients hospitalized with AHF, therefore, our objective is to address the current knowledge gap surrounding the effectiveness of SGLT2 inhibitors compared with placebo in patients with or without type 2 diabetes. Applying different endpoint measures, our primary outcome comprises estimating risks of all-cause and cardiovascular mortality, heart failure events, readmissions, and symptomatic improvement. Our secondary outcome is the risk evaluation of serious adverse events.

## 2. Methods

### 2.1. Study Design and Strategy

This study has been designed in accordance with the Preferred Reporting Items for Systematic Reviews and Meta-Analyses (PRISMA) Statement 2020 guidelines [23]. This meta-analysis was registered with PROSPERO 2022 CRD42022365431. A comprehensive literature search of digital databases, including PubMed, Scopus, and Cochrane Library was performed from inception to August 13, 2022. Keywords and medical subject headings (MESH) were combined and run on the PubMed database with the following terms: “sglt2,” “sodium-glucose transporter 2 inhibitors,” “sglt2 inhibitors,” “decompensated heart failure,” “de-novo heart failure,” “acute heart failure”. The Boolean (and/or) logic was applied. Search filters were not applied to ensure maximal, relevant publications were obtained. This strategy was tailored to other databases including Scopus, and Cochrane Library. The reference lists were also reviewed for review articles to identify all relevant studies (umbrella methodology). Potential unpublished studies on ClinicalTrials.gov were also reviewed to locate any ongoing trials and include any relevant data.

### 2.2. Eligibility Criteria

All RCTs, including patients aged > 18 years, hospitalized with de-novo acute heart failure (AHF) or acutely decompensated chronic heart failure with reduced, borderline, or preserved ejection fraction, and intervention group receiving SGLT2 inhibitors that have been approved or are currently investigational were included. All observational studies, letters to the editor, single-arm studies, and two-arm studies that did not report any outcomes of interest were excluded. There was no restriction placed with respect to the diagnosis of type 2 diabetes or lack thereof. Outcome measures were decided a priori based on already-published data on heart failure (HF) patients who received SGLT2 inhibitors for non-AHF conditions. Efficacy outcomes included all-cause mortality, patients with heart failure events, total heart failure events, cardiovascular mortality, readmissions, and change in Kansas City Cardiomyopathy Questionnaire (KCCQ) Total Symptom Score. Safety outcomes included serious adverse events, acute kidney injury, hepatic injury, hypotension, hypoglycemia, urinary tract infections (UTIs), and diabetic ketoacidosis (DKA).

### 2.3. Study Selection and Data Extraction

Two investigators (A.S. and Z.S.) independently conducted the screening and finalization of the articles. The titles and abstracts were screened in the first stage by each investigator independently. Any discrepancies were resolved by a third investigator (I.C.O.) in case of any disagreements. Shortlisted articles were screened for full-text eligibility by the two investigators independently. In the final stage, the discrepancies were solved by the third investigator, and the studies were included in the review. Data were systematically collected for efficacy and safety outcomes. For all identified studies, the findings were tabulated as (i) clinical trial identifier, (ii) author-year, (iii) country, (iv) dose of SGLT2, (v) frequency of intervention, (vi) total sample size, (vii) intervention arm sample size, (viii) control arm sample size, (ix) follow-up duration, (x) primary outcome, and (xi) secondary outcome.

### 2.4. Statistical Analysis

All studies identified from the databases were stored in Endnote X9 (Clarivate Analytics, London, UK). The duplicates were removed using the Endnote X9 deduplication tool. No duplicates were found using the online resources (i.e., ClinicalTrials.Gov). The methodology was both quantitative and analytical to ascertain the benefit of SGLT2 in reducing the risk of adverse outcomes following hospitalization for AHF. The KCCQ scale scores were continuous, and the difference in means along with standard deviation was computed. On noting these values, the standardized mean difference (SMD) was computed, reported as Cohen’s d, applying 95% confidence intervals (CI). A random-effects model was applied as it was assumed that the observed estimates of treatment effect can vary across studies because of real differences in the treatment effect in each study as well as sampling variability (chance). Forest plots were generated for every outcome documenting Risk Ratio (RR), SMD, 95% CI, heterogeneity, and overall results. The formula for RR and SMD is as follows:$$\mathrm{RR}=\frac{risk\;of\;event\;in\;the\;SGLT2\;group}{risk\;of\;event\;in\;the\;control\;group}\;\mathrm{SMD}=\frac{difference\;in\;mean\;outcome\;between\;groups}{standard\;deviation\;of\;outcome\;among\;participants}$$

The minimum requirement to meta-analyze the findings was two or more studies reporting the same outcome measure. While a funnel plot was not generated to test for publication bias due to the limited number of studies (<10), the heterogeneity between the included studies was tested using the χ^2^-based Q test and the I^2^ index. The statistical analysis was conducted using Review Manager 5.4 (RevMan, Cochrane).

### 2.5. Risk of Bias Assessment

The risk of bias was assessed for the included studies individually by two investigators (A.S. and Z.S.). The risk-of-bias tool for randomized trials (RoB 2) version 2 available in the Cochrane Handbook for Systematic Reviews of Interventions was utilized [24]. Five fixed sets of domains of bias were graded as “low,” “high,” or “some concerns.” The five domains comprised: (i) bias from the randomization process, (ii) bias due to deviations from intended interventions, (iii) bias due to missing outcome data, (iv) bias in the measurement of the outcome, and (v) bias in the selection of reported results. In case of discrepancies between the two investigators, the third investigator (I.C.O.) resolved it through consensus.

## 3. Results

### 3.1. PRISMA

We identified 2627 articles, and 1985 titles and abstracts were screened after duplicates were removed. There were 529 articles that met the criteria for full-text review and 3 articles were included. The trial selection process is presented in a PRISMA flow diagram (Figure 1). There was a strong agreement among the reviewers for the inclusion of the articles; the title and abstract screening stage yielded a score of: κ = 0.86; and for full-text screening (κ = 0.88).

### 3.2. Risk of Bias

Each included study was critically appraised using the revised tool for Risk of Bias (ROB 2) for randomized trials at the level of five outcomes: all-cause mortality, heart-failure events, re-admissions, symptom score, and adverse events (Figure 2). All studies were considered to have a “low” risk of bias based on the five pre-specified domains in the ROB-2 tool. One trial was identified as having “some concerns” due to deviation in reporting pre-specified primary outcomes that may introduce bias in the selection of the reported result (Domain 5).

### 3.3. Baseline Characteristics

A total of 1831 patients were included of whom 913 received intervention (49.9%). The drug of choice was Empagliflozin in EMPULSE [25] and EMPA-RESPONSE-WHF [26] and Sotagiflozin in SOLOIST-WHF [27]. Baseline characteristics were mostly similar between interventional and control groups. The overall mean age in the interventional arm was 72.9 (SD: 5.3) years. In the control arm, the overall mean age was 70.6 (SD: 2.1) years. Only 33.7% (*n* = 617) of the total sample was female. Follow-up duration ranged from 60 days to 9 months. A total of 770 (84.3%) of the patients in the intervention arm and 758 (82.6%) of the patients in the control arm had diabetes as a comorbid. Key characteristics are summarized in Table 1.

### 3.4. All-Cause Mortality

All three trials reported the data on all-cause mortality at different time points. Compared with the placebo group, the risk of mortality was reduced by 27% in the intervention group (RR: 0.73, 95% CI: 0.49–1.09, *p* = 0.12, I^2^ = 18%) (Figure 3). For the sensitivity test, SOLOIST-WHF [27] had the highest weight and was thus removed to determine whether there will be changes in the risk. The post-sensitivity analysis was conducted to further explore the risk after the SOLOIST-WHF trial was removed which had the highest weight [27]. It demonstrated an even larger reduction in risk of death such that the group receiving SGLT2 inhibitors still had a lowered risk of all-cause mortality at 51% compared to the placebo group, which was significant (RR: 0.49, 95% CI: 0.25–0.95, *p* = 0.04, I^2^ = 0%). Both EMPA-RESPONSE-AHF [26] and SOLOIST-WHF [27] were analyzed separately as patients with acute decompensated chronic HF (ADCHF) were included in both these trials; the mortality risk reduction was 15% in patients with ADCHF who took SGLT2 inhibitors compared to placebo (RR:0.85, 95% CI: 0.62–1.15, *p* = 0.39, I^2^ = 0%).

### 3.5. Heart Failure Events

Heart failure events (HFEs) were defined as a hospitalization/ER visit, an urgent care visit, or an outpatient visit requiring intensification of management (intravenous therapy, mechanical support, renal, or circulatory support) for worsening signs and/or symptoms of heart failure. All three trials reported heart failure events (HFEs) among the participants. Compared to the placebo group, the intervention group had a significant risk reduction of 62% in HFEs (RR: 0.66, 95% CI: 0.58–0.75, *p* < 0.0001, I^2^ = 0%) (Figure 4A). After removing the SOLOIST-WHF trial [27], the sensitivity analysis still indicated a 45% reduction in HFEs in favor of the intervention group which did not reach statistical significance (RR: 0.55, 95% CI: 0.24–1.26, *p* = 0.16, I^2^ = 48%). The total number of HFEs presented was identified during the entire duration of the study. Two of three trials reported the total number of HFEs across participants. There was a significant risk reduction of 35% in the number of HFEs across the trials (RR: 0.65, 95% CI: 0.46–0.91, *p* = 0.01, I^2^ = 0) (Figure 4B). The total number of patients who had worsening heart failure or had died is attributable to cardiovascular causes till the end-of-trial visit was included. Two trials reported a combined endpoint attributable to the total number of patients who died of cardiovascular causes or had HFEs. There was a significant reduction of 28% in the intervention group for risk of either cardiovascular death or HFEs (RR: 0.82, 95% CI: 0.73–0.93, *p* = 0.002, I^2^ = 0%) (Figure 4C).

### 3.6. Readmissions

All three trials reported readmissions due to HFEs. The endpoints for readmissions in EMPULSE [25] and EMPA-RESPONSE-AHF [26] were 30 and 60 days, respectively. The SOLOIST-WHF [27] reported readmission rates during the entire duration of the study. There was a 24% risk reduction of first-time readmission among the intervention group who had been discharged following acute heart failure, compared to the placebo group, which was significant (RR: 0.76, 95% CI: 0.60–0.98, *p* = 0.03, I^2^ = 4%) (Figure 5). A sensitivity analysis was conducted and SOLOIST-WHF [27] was removed to identify the readmission risk within 60 days as reported in EMPULSE [25] and EMPA-RESPONSE-AHF [26]. The readmission risk was reduced by 15% which did not receive statistical significance due to the smaller sample size of the remaining two trials (RR: 0.85, 95% CI: 0.31–2.31, *p* = 0.75, I^2^ = 36%).

### 3.7. Kansas City Cardiomyopathy Questionnaire Total Symptom Score (KCCQ TSS)

Two of three trials noted the adjusted mean reduction in the Kansas City Cardiomyopathy Questionnaire Total Symptom Score (KCCQ TSS). An analysis was conducted to note the effect size of intervention vs. placebo for KCCQ TSS. While a small effect size in favor of intervention was found, the KCCQ-TSS reduction was significant which indicates improvement in the group receiving SGLT2 inhibitors (Cohen’s d = −0.27, 95% CI = −0.40, −0.14, *p* < 0.0001, I^2^ = 41%) (Figure 6).

### 3.8. Serious Adverse Events

All three trials reported safety outcomes of participants receiving either intervention or placebo. The risk for serious adverse events was slightly lowered in the intervention group by 15% which was insignificant, demonstrating no safety concerns in the three trials concerning SGLT2 inhibitor use in acute heart failure (RR: 0.85, 95% CI: 0.70–1.03, *p* = 0.1, I^2^ = 44%) (Figure 7). Compared to the placebo, there was a 22% reduced risk of developing acute kidney injury with SGLT2 inhibitors yet did not reach statistical significance (RR: 0.78, 95% CI: 0.54–1.14, *p* = 0.2 I^2^ = 0%) (Figure 8). There was a comparative risk reduction of 23% of developing hepatic injury with SGLT2 inhibitors compared to placebo yet did not reach statistical significance (RR: 0.77, 95% CI: 0.38–1.55, *p* = 0.46, I^2^ = 0%) (Figure 9). There was a comparative increase in risk by 18% for the development of hypotension with SGLT2 inhibitors compared to placebo yet did not reach statistical significance (RR: 1.18, 95% CI: 0.76–1.84, *p* = 0.45, I^2^ = 0%) (Figure 10). There was a relatively increased risk of hypoglycemia by 49% with SGLT2 inhibitors compared to placebo yet did not reach statistical significance (RR: 1.49, 95% CI: 0.86–2.58, *p* = 0.79, I^2^ = 0%) (Figure 11). Compared to the placebo, there was a 16% reduced risk of developing urinary tract infections with SGLT2 inhibitors yet did not reach statistical significance (RR: 0.84, 95% CI: 0.56–1.27, *p* = 0.41, I^2^ = 0%) (Figure 12). There was a notable risk reduction of 54% in the group receiving SGLT2 inhibitors when compared to placebo but did not reach statistical significance (RR: 0.46, 95% CI: 0.10–2.04, *p* = 0.31, I^2^ = 0%) (Figure 13).

### 3.9. Ongoing Clinical Trials of SGLT2 inhibitors and Heart Failure

As of 17th November 2022, 15 ongoing clinical trials are being conducted testing the efficacy and safety of SGLT2 inhibitors on heart failure, diabetes mellites type 2, acute myocardial infarction, and chronic kidney disease (Table 2). The interventions consist of Empagliflozin (10 mg, 25 mg), Dapagliflozin (10 mg), and Canagliflozin. Control and comparator group standard care approaches consist of either placebo or loop diuretics, vasodilators, inotropic agents, digoxin, and/or vasopressors.

With a total enrollment of 17,503 participants, the ongoing trials are being conducted worldwide in 27 countries: Argentina, Australia, Brazil, Bulgaria, Canada, China, Denmark, Egypt, France, Germany, Hong Kong, Hungary, India, Israel, Japan, Japan, Korea, Mexico, Netherlands, Poland, Romania, Russian Federation, Serbia, Sweden, United Kingdom, Switzerland, Ukraine, and the United States. The outcome measures, phase of study, enrollment, and completion date for ongoing trials are enlisted in Table 3.

## 4. Discussion

The primary objective of our systematic review and meta-analysis was to examine the efficacy of SGLT2 inhibitors compared to placebo for reduction of all-cause and cardiovascular mortality, heart failure events (HFEs), readmissions, and symptomatic improvement in patients admitted for acute heart failure. Our secondary objective was to assess the risk of serious adverse events in patients admitted for acute heart failure, receiving either SGLT2 inhibitors or a placebo. Our meta-analysis compiled the findings of three studies that utilized SGLT2 inhibitors in either acutely decompensated chronic HF (ADCHF) or de novo acute HF (AHF). We found that SGLT2 inhibitors reduced the risk of all-cause mortality by 27% and were significant at 51% when a post-sensitivity analysis was conducted. Furthermore, we also found a significant reduction in cardiovascular mortality and HFEs by 28% among patients receiving SGLT2 inhibitors. Specifically, the absolute number of HFEs and total patients with HFEs were significantly reduced by 62% and 35% among those who used SGLT2 inhibitors, respectively. Moreover, we found a significant reduction in first-time re-admission rates by 24% among patients with SGLT2 inhibitors. These findings were supported by significant symptomatic improvement and no additional safety concerns [28].

The initial evidence of the efficacy of SGLT2 inhibitors in the clinical trial setting was noted in EMPA-REG-OUTCOME [19]—where it was determined that individuals hospitalized for heart failure, and who were randomized into treatment groups had a two-fold reduced risk of being re-hospitalized or dying in the first 1–3 months of the first heart failure event [19]. The EMPA-RESPONSE-AHF trial was the first prospective clinical trial to have evaluated the clinical benefits of SGLT2 inhibitors among acute heart failure patients [26]. This trial was a double-blind, randomized, placebo-based, parallel-group study based in multiple centers; 80 individuals with acute heart failure with or without type 2 diabetes mellitus were randomized to either 10 mg/day of empagliflozin or to the control group; all randomized and medicinal administration was pursued within 24 h of admission [26]. The EMPULSE (EMPagliflozin in patients hospitalized with acUte heart faiLure who have been StabilizEd) trial was based in multiple countries/centers, randomized with a double-blind setting where the effects of the SGLT2 inhibitor (empagliflozin) was assessed for safety, clinical benefit and tolerability for acute heart failure [25,29]. In this trial, patients underwent an initial stabilization period which had a median time span of 3 days. The patients also received 10 mg per day of the SGLT2 inhibitor (empagliflozin) or no intervention with standard care for 90 days. The EMPULSE trial met its endpoint, where patients showed more clinical benefits as compared to placebo; the stratified win ratio was 1.36 (95% CI = 1.09 to 1.68, *p* = 0.0054) [25]. Notably, the benefits were unanimous throughout the different subgroups including those that had decompensated chronic heart failure and ventricular ejection fractions upwards or downwards of 40%. The intervention was considered to be both well-tolerated and safe for the patients.

Other representatives of the class including dapagliflozin are currently being investigated. The DISTATE-AHF, which is a multicenter, prospective, open-label, randomized trial is enrolling 240 patients in the US [30]. The patient population consists of patients with type 2 diabetes mellitus hospitalized with hypervolemic AHF and glomerular filtration rate above 30 mL/min/1.73m^2^ [30]. With endpoints consisting of diuretic response, inpatient AHF, 30-day readmission rate, and safety metrics, it is yet to be quantified whether dapagliflozin will be a candidate therapeutic for AHF among patients with diabetes [30]. In the CHIEF-HF trial, 467 participants with heart failure regardless of diabetes or ejection fraction status were randomized to canagliflozin (100 mg) or placebo intervention [31]. While the enrollment was stopped early because of sponsor priorities, Spertus and colleagues (2022) report that the primary outcome of KCCQ TSS was changed at 12 weeks by 4.3 points (*p* = 0.016)—favoring canagliflozin [31]. The CHIEF-HF met its primary endpoint; with a shift in the paradigm of care for heart failure imminent [31], it is important to further test and review the findings of SGLT2 inhibitors in randomized, double-blind settings.

In our meta-analysis, we ascertain the pooled findings of all three RCTs published in this arena so far. The EMPULSE trial works as an add-on/complement to the results of the previous two trials that administered SGLT2 inhibitors; these trials depict the patient journey from undergoing acute heart failure to in-hospital admission and use of the intervention. Our findings are noteworthy as we collate proof to showcase that SGLT2 inhibitors can be considered an element of usual care, translating to clinical benefit [32]. Our study findings support that SGLT2 inhibitors help manage AHF patients following hospitalization.

### 4.1. Clinical Practice Recommendations

With abundant data on the use of SGLT2 inhibitors for heart failure, it is essential to review the feasibility in clinical practice. So far, literature reports that SGLT2 inhibition interacts with key pathways at the cellular level, thereby providing cardioprotective effects to patient populations. The mechanisms include cardiac remodeling, myocardial calcium handling, lipolysis, and modification of the epicardial thickness [33].

SGLT2 inhibitors reduce the risk of composite and specific heart failure outcomes. The benefits of SGLT2 inhibitor use are large among those with a history of heart failure and with diabetes. Moreover, benefits are also noted in weight control, blood pressure regulation, and hemoglobin levels. It may be worthwhile to consider SGLT2 inhibitor treatment initiation if heart failure predominates since the intervention is an excellent target for blood glucose and control. No notable adverse events are reported among patients with heart failure being treated with SGLT2 inhibitors as compared to the standard of care. The benefits of treatment on heart failure outcomes are insofar high based on current evidence when compared to other therapies used for heart failure management including ACE inhibitors.

### 4.2. Limitations

Our meta-analysis has certain limitations that must be reported. First, the duration of the intervention was variable across all trials. Second, there was limited uniformity in how the trials defined HFEs. Third, the follow-up duration was inconsistent. Certain aspects of care were administered for longer periods across the trials. Fourth, the outcome measures were conducted at different time points e.g., readmission rates. Fifth, the sample size was fairly small, with SGLT2 inhibitor group enrolled participants not reaching generalizability. Lastly, a majority of the sample was diabetic, which may lead to biases when making broadly characterizable decisions for patient care.

## 5. Conclusions

The current study looked into the efficacy and safety of SGLT2 inhibitors in preventing complications following AHF. By compiling the findings of three trials, we analyzed a total sample of 1831 patients irrespective of their diabetes status. Moreover, we also qualitatively critiqued 15 ongoing clinical trials administering SGLT2 inhibitors across 27 countries for a variety of cardiovascular conditions including acute heart failure. We found that SGLT2 inhibitors decrease the odds of all-cause mortality, cardiovascular mortality, heart failure events, and re-admission rates within the first 1–9 months of hospitalization. Additional studies are necessary to fully understand the beneficial impact of SGLT2 inhibitors in managing AHF patients without diabetes.

## Figures and Tables

**Figure 1 healthcare-10-02356-f001:**
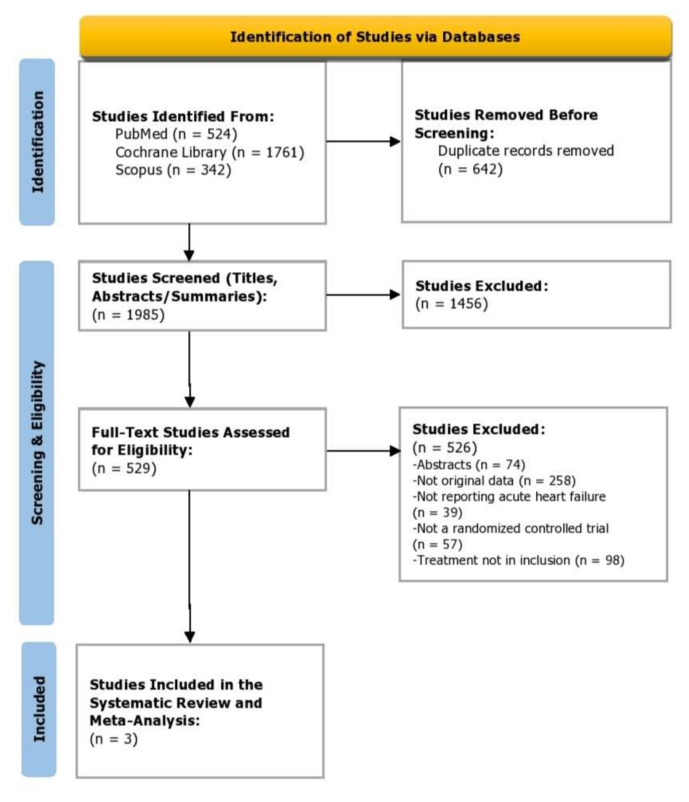
PRISMA flow diagram depicting study selection.

**Figure 2 healthcare-10-02356-f002:**
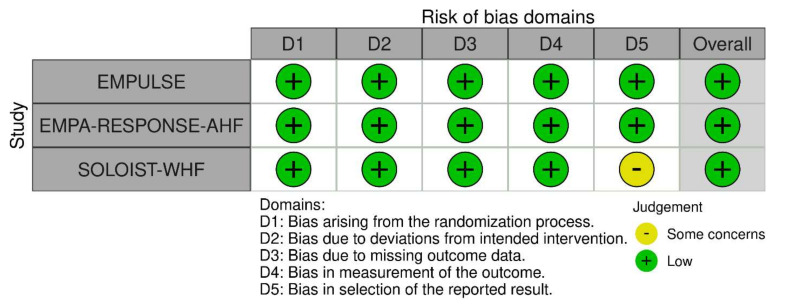
Risk of bias assessment using the Risk of Bias 2.0 tool [25,26,27].

**Figure 3 healthcare-10-02356-f003:**
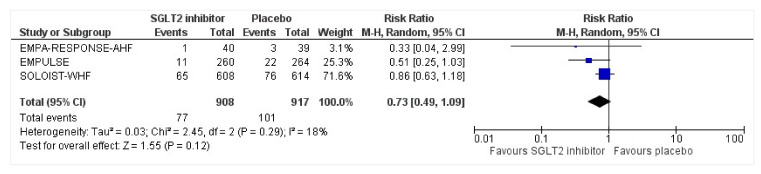
Forest plot for all-cause mortality of patients with acute heart failure receiving sodium-glucose cotransporter 2 inhibitor or placebo. The total number of events is presented in the given study duration [25,26,27]. CI: confidence interval; M-H: Mantel–Haenszel; SGLT2: sodium-glucose cotransporter 2.

**Figure 4 healthcare-10-02356-f004:**
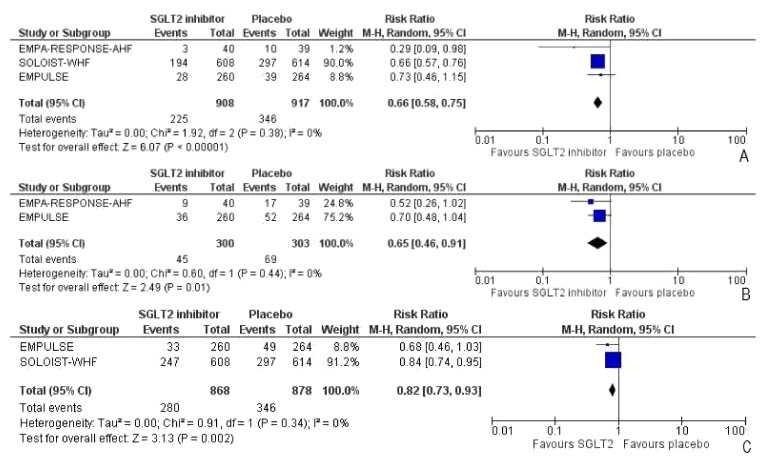
Forest plot for (**A**) the number of participants with heart failure events (the total number of participants presented who had one or more HFEs in the duration of the study), (**B**) the number of heart failure events, and (**C**) the number of patients with either HFEs or death attributable to cardiovascular causes [25,26,27].

**Figure 5 healthcare-10-02356-f005:**
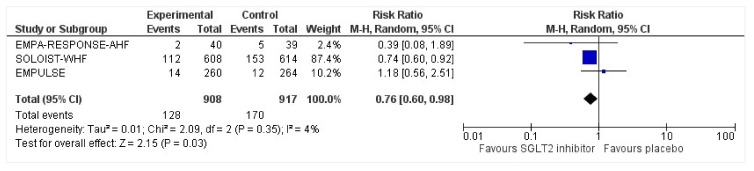
Forest plot for the number of patients requiring readmission following discharge for acute heart failure in either group [25,26,27].

**Figure 6 healthcare-10-02356-f006:**
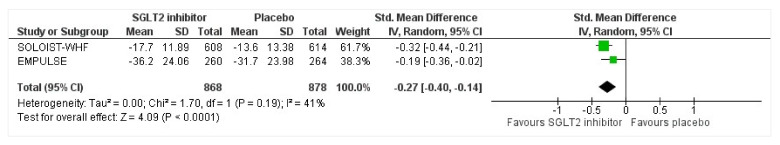
Forest plot for the adjusted mean change in KCCQ-TSS (mean values [SD]) at 90 days from baseline in either group [25,27].

**Figure 7 healthcare-10-02356-f007:**
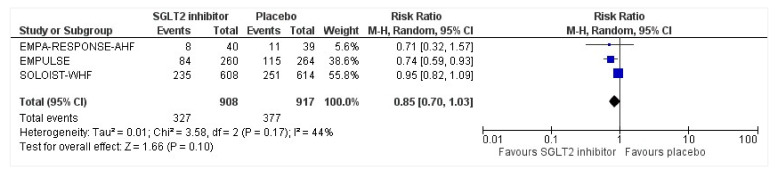
Forest plot of the total number of patients who had at least one serious adverse event in either group [25,26,27].

**Figure 8 healthcare-10-02356-f008:**
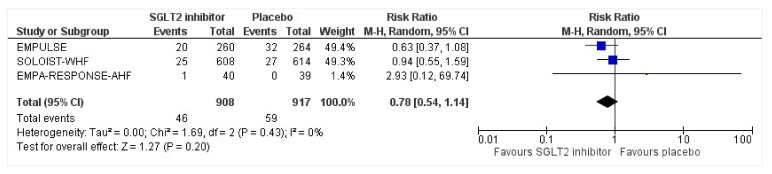
Forest plot for the total number of patients who had acute renal failure as a serious adverse event during the duration of treatment in either group [25,26,27].

**Figure 9 healthcare-10-02356-f009:**
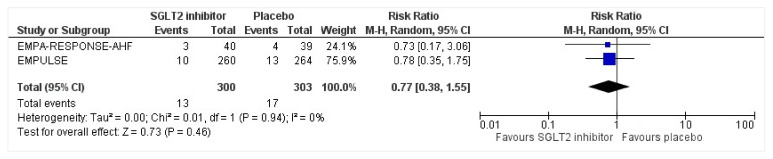
Forest plot for the total number of patients who had a hepatic injury as a serious adverse event during the duration of treatment in either group [25,26].

**Figure 10 healthcare-10-02356-f010:**
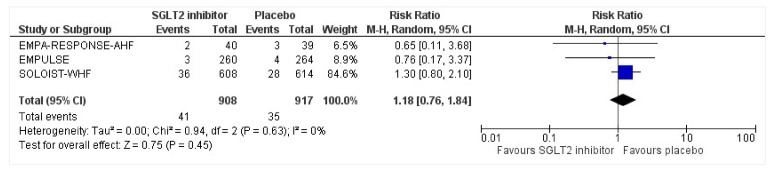
Forest plot for the total number of patients who had hypotension as a serious adverse event in either group [25,26,27].

**Figure 11 healthcare-10-02356-f011:**
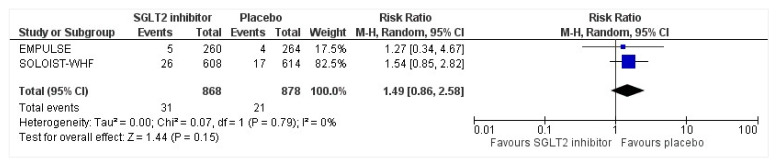
Forest plot for the total number of patients who had hypoglycemia as a serious adverse event in either group [25,27].

**Figure 12 healthcare-10-02356-f012:**
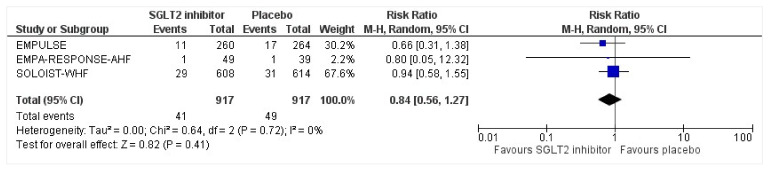
Forest plot for the total number of patients who had a urinary tract infection as a serious adverse event in either group [25,26,27].

**Figure 13 healthcare-10-02356-f013:**
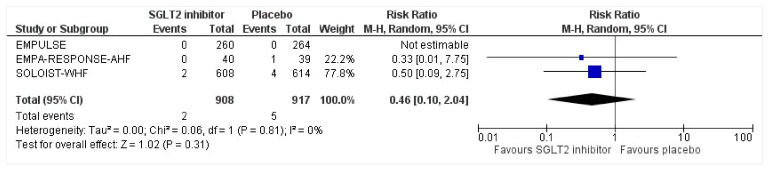
Forest plot for the total number of patients who had DKA as a serious adverse event in either group [25,26,27].

**Table 1 healthcare-10-02356-t001:** Trial characteristics of the included studies. Only patients who had acute heart failure with or without diabetes were included in the trials.

	EMPULSE	EMPA-RESPONSE-AHF	SOLOIST-WHF
Clinical trial identifier	NCT04157751	NCT03200860	NCT03521934
Author-year	Voors-2022 [25]	Damman-2020 [26]	Bhatt-2021 [27]
Country	The Netherlands	The Netherlands	United States of America
Dose of SGLT2	10 mg Empagliflozin	10 mg Empagliflozin	200–400 mg Sotagliflozin
Frequency of intervention	Every day for 90 days	Every day for 30 days	Every day for 8 months with uptritation depending on side effects
Total sample size	530	79	1222
Intervention arm sample size	265	40	608
Control arm sample size	265	39	614
Follow-up duration	90 days	60 days	9 months
Primary outcome(s)	Clinical benefit is defined as a hierarchical composite of all-cause mortality, number of heart failure events, and time to a first heart failure event, or a 5-point or more difference in baseline score of KCCQ-TSS	Change in visual analog scale (VAS) dyspnea score, diuretic response, the percentage change in N-terminal pro-brain natriuretic peptide (NT-proBNP), and length of hospital stay	Total number of deaths from cardiovascular causes and hospitalizations and urgent visits for heart failure
Secondary outcome(s)	Cardiovascular death or HFE, KCCQ-TSS improvement of 10 or more points, reduction in NT-proBNP concentration, hospitalization for HF, and diuretic response	Worsening HF, all-cause mortality, and HF readmission	Number of hospitalizations and urgent visits for HF, cardiovascular death, all-cause mortality, and change in KCCQ-TSS score

**Table 2 healthcare-10-02356-t002:** Ongoing clinical trial characteristics of SGLT2 inhibitors and varying heart failure conditions.

Sr. No.	NCT Number	Title	Acronym	Status	Conditions	Interventions
1	NCT04298229	Efficacy and Safety of Dapagliflozin in Acute Heart Failure	DICTATE-AHF	Recruiting	Heart Failure, Diabetes Mellitus Type 2	Dapagliflozin 10 mg; protocolized diuretic therapy
2	NCT05392764	Early Treatment with a Sodium-glucose Co-transporter 2 Inhibitor in High-risk Patients With Acute Heart Failure	EMPA-AHF	Recruiting	Acute Heart Failure	Empagliflozin 10 mg; placebo
3	NCT05556044	Empagliflozin for New On-set Heart Failure Study Regardless of Ejection Fraction	EMPA	Recruiting	Acute Heart Failure	Empagliflozin 10 mg
4	NCT05406505	Effect of Dapagliflozin in Patients With Acute Heart Failure (DAPA-RESPONSE-AHF)		Recruiting	Acute Decompensated Heart Failure	Dapagliflozin 10 mg; placebo
5	NCT04899479	Peri-treatment of SGLT-2 Inhibitor on Myocardial Infarct Size and Remodeling Index in Patients With Acute Myocardial Infarction and High Risk of Heart Failure Undergoing Percutaneous Coronary Intervention	PRESTIGE-AMI	Recruiting	Acute Myocardial Infarction, Heart Failure	SGLT2 inhibitor; control
6	NCT05346653	The Hemodynamic Effects of SGLT2i in Acute Decompensated Heart Failure		Not yet recruiting	Acute Decompensated Heart Failure	SGLT2 inhibitor; control
7	NCT05305495	Empagliflozin in Acute Heart Failure	DRIP-AHF-1	Not yet recruiting	Acute Heart Failure, Chronic Kidney Diseases	Empagliflozin 25 mg
8	NCT04363697	Dapagliflozin and Effect on Cardiovascular Events in Acute Heart Failure -Thrombolysis in Myocardial Infarction 68 (DAPA ACT HF-TIMI 68)		Recruiting	Acute Heart Failure, Heart Failure	Dapagliflozi; placebo
9	NCT04782245	Acute Reno-Cardiac Action of Dapagliflozin In Advanced Heart Failure Patients on Heart Transplant Waiting List	ARCADIA-HF	Not yet recruiting	End-stage Heart Failure	Dapagliflozin 10mg; placebo
10	NCT04869124	Dapagliflozin on Volume Vascular Outcomes.	DAPA-VOLVO	Recruiting	Heart Failure, Congestive	Dapagliflozin; placebo
11	NCT04778787	Sodium-glucose Cotransporter Type 2 Inhibitors for Acute Cardiorenal Syndrome Prevention		Recruiting	Congestive Heart Failure	Standard list of drugs used for acute decompensation of CHF (loop diuretics, vasodilators, digoxin, inotropic agents, vasopressors), plus dapagliflozin (Forxiga; MP-002596)
12	NCT04717986	Dapagliflozin Effects on Mayor Adverse Cardiovascular Events in Patients with Acute Myocardial Infarction (DAPA-AMI)	DAPA-AMI	Enrolling by invitation	Acute Myocardial Infarction, Cardiovascular Morbidity, Heart Failure, Angina, Unstable	Dapagliflozin 10 mg; placebo
13	NCT04564742	Dapagliflozin Effects on Cardiovascular Events in Patients with an Acute Heart Attack	DAPA-MI	Recruiting	Acute Myocardial Infarction, Heart Failure	Dapagliflozin; placebo
14	NCT04509674	EMPACT-MI: A Study to Test Whether Empagliflozin Can Lower the Risk of Heart Failure and Death in People Who Had a Heart Attack (Myocardial Infarction)		Recruiting	Myocardial Infarction	Empagliflozin; placebo
15	NCT05364190	Canagliflozin in Patients with Acute Decompensated Heart Failure		Recruiting	Chronic Heart Failure, Acute Decompensated Heart Failure, Diabetes Mellitus	Canagliflozin and Empagliflozin

**Table 3 healthcare-10-02356-t003:** Ongoing clinical trial characteristics of outcome measures, phase of study, enrollment, completion date, and locations.

Sr. No.	NCT Number	Outcome Measures	Phases	Enrollment	Completion Date	Locations
1	NCT04298229	Cumulative change in weight (kilograms); Incidence of worsening heart failure; Hospital readmission	Phase 3	240	31 January-2023	United States
2	NCT05392764	Within 90 days: Composite endpoint consisting of death, heart failure rehospitalization; WHF during hospitalization, urine output up to 48 h after treatment initiation; worsening NYHA class; Improvement in KCCQ-TSS points from randomization to 30 and 90 days after treatment initiation; Time to hemodynamic stabilization during index hospitalization; Death; Composite of renal replacement therapy, renal transplantation, eGFR <15 mL/min/1.73m^2^; Trend in eGFR after randomization to 24 h, 48 h, 30 days, and 90 days	Phase 3	500	1 April 2023	Japan
3	NCT05556044	Heart failure (HF) events; All-cause mortality; KCCQ-TSS total symptom score; NT-proBNP level; New York Heart Association (NYHA) class; Major Adverse Cardiovascular Event (MACE); Occurrence of kidney damage; Weight loss; Quality-adjusted life years (QALY) gained; Change in 6 min hall walk (6MHW)	Phase 3	200	31 May 2024	Hong Kong
4	NCT05406505	Change in dyspnea- Visual analog scale; Incidence of worsening heart failure (HF); All-cause death; Hospital readmission; Urinary sodium 2 h post randomization; Difference in serum levels of congestion biomarkers	Phase 2/3	100	5 January 2023	Egypt
5	NCT04899479	Myocardial infarct size; Left ventricular end-systolic volume; Acute kidney injury; Myocardial salvage index (MSI); Microvascular obstruction (MVO); Hemorraghic infarction (HI); Thrombolysis in myocardial infarction (TIMI) flow grade; ST resolution after PCI; left ventricular end-diastolic volume; left ventricular ejection fraction; LV adverse remodeling; LV reverse remodeling; MSI; MVO; Changes of NT-proBNP level; Estimated glomerular filtration rate; Cardiac death or re-hospitalization due to heart failure; All-cause death or re-hospitalization due to heart failure; Target lesion failure; Target vessel failure; All-cause death; Cardiac death; Target vessel; Re-hospitalization due to heart failure; Any re-hospitalization	Phase 4	200	30 June 2024	Republic of Korea
6	NCT05346653	Change in Indirect Fick cardiac index; Change in pulmonary capillary wedge pressure (PCWP)	Phase 4	40	October 2023	United States
7	NCT05305495	The diuretic effect of empagliflozin in association with furosemide; Fractional excretion of sodium in the urine; Total urine sodium output; Changes in volume status; Incidence of AKI; Electrolyte abnormalities—Sodium; Electrolyte abnormalities- Potassium; Electrolyte abnormalities—Magnesium	Phase 4	25	July 2025	Canada
8	NCT04363697	Cardiovascular (CV) death or worsening heart failure; Composite CV death, rehospitalization for heart failure, urgent heart failure visit; Composite CV death, rehospitalization for heart failure; Rehospitalization for heart failure, urgent heart failure visit; Readmission; CV death; Death	Phase 4	2400	31 May 2023	United States
9	NCT04782245	Levels of suPAR (ng/mL); VO2 max assessment; assessed by right heart catheterization: cardiac output, pulmonary capillary wedge pressure, pulmonary artery systolic and mean pressure, mean pressure, right atrial pressure; assessed by echocardiograpgy: left ventricular ejection function, left ventricular end-diastolic diameter, left ventricular end-systolic volume, mitral regurgitation grade, left atrial volume; Nt-proBNP level; Creatinine level; Quality of life assessed by KCCQ	Phase 2	80	April 2024	France
10	NCT04869124	Change in relative plasma volume status, blood volume, red blood cell volume, total hemoglobin mass, extracellular to total body water ratio, intracellular to total body water ratio, flicker-light induced retinal arteriolar dilatation, flicker-light induced retinal venular dilatation, retinal arterial to venous ratio, pulse wave velocity, flow-mediated dilatation of the brachial artery, glyceryl-trinitrate- induced dilatation of the brachial artery	Phase 4	80	31 December 23	Switzerland
11	NCT04778787	Death due to heart failure; deterioration of renal function (increase in blood creatinine by 0.3 mg/dl within 48 h); development of resistance to diuretics; re-hospitalization about decompensation of chronic heart failure within 30 days after discharge from the hospital	Phase 4	370	1 August 2022	Russian Federation
12	NCT04717986	Mayor adverse cardiovascular effects; Left ventricular ejection fraction; Chronic heart failure; Post infarction angina; Mortality due to cardiovascular cause	NA	188	1 September 2022	Mexico
13	NCT04564742	Time to the first occurrence of any of the components of this composite: hospitalization for heart failure or cardiovascular death; Time to the first occurrence of any of the components of this composite: myocardial infarction or stroke (incl. ischaemic, hemorrhagic, and undetermined stroke) or cardiovascular death; Time to the first occurrence of a fatal or a non-fatal MI/CV Death/death of any cause/new onset of type 2 diabetes mellitus post-randomization	Phase 3	6400	22 September 2023	Sweden, United Kingdom
14	NCT04509674	Composite of time to first heart failure hospitalization or all-cause mortality; Total number of HHF or all-cause mortality; Total number of non-elective Cardiovascular (CV) hospitalizations or all-cause mortality; Total number of non-elective all-cause hospitalizations or all-cause mortality; Total number of hospitalizations for MI or all-cause mortality; Time to CV mortality	Phase 3	6500	31 March 2023	United States, Argentina, Australia, Brazil, Bulgaria, Canada, China, Denmark, France, Germany, Hungary, India, Israel, Japan, Korea, Netherlands, Poland, Romania, Russian Federation, Serbia, Spain, Ukraine
15	NCT05364190	The cumulative mean of daily diuresis, diuretic; change in the level of NT-pro BNP; Presence of symptoms of congestion and dyspnea at discharge; ICU length of stay; incidence of worsening of heart failure case; fractional excretion of sodium-based diuretic efficiency; Serum potassium; Incidence of ketoacidosis; Serum glucose covariate-adjusted for baseline with attention to both elevations; Incidence of symptomatic, sustained hypovolemic hypotension; In-hospital mortality; Hospital readmission within 90 days of discharge for heart failure; Incidence of mortality within 90 days from discharge due cardiovascular cause; The incidence of worsening of renal function; Any reported adverse events during follow up period; The progression of heart failure severity	Phase 3	180	9 November 2023	Egypt

## Data Availability

All data utilized for the purpose of this study are available publicly and online.

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
