# Peer review of "SGLT2 Inhibitors in Acute Heart Failure: A Meta-Analysis of Randomized Controlled Trials"

_healthcare, 2022, doi:10.3390/healthcare10122356_

Round 1
Reviewer 1 Report
Regarding the manuscript "SGLT2 Inhibitors in Acute Heart Failure: A Meta-Analysis of 2 Randomized Controlled Trials" the following issues should be mentioned:
1. The list of keyword could be limited to those terms investigates in the manuscript
2. The Figures could be enlarged for better clarity (1, 3-13)
3. The Discussion chapter could be enriched with data regarding other representatives of the class: dapagliflozin (e.g. ongoing DICTATE-AHF trial) and canagliflozin.
After a deeper look in the current published medical literature, I found a recent meta-analysis published on the exact topic that included the same 3 studies, with exactly the same number of patients: https://cardiab.biomedcentral.com/articles/10.1186/s12933-022-01455-2. The present manuscript lack originality and should be improved by adding newer data that will be available in the near future when DICTATE-AHF trial or other trials will be published. Therefore, I will change my recommendation to Reject the present manuscript.
Author Response
Reviewer 1 Comments and Author Responses:
Regarding the manuscript "SGLT2 Inhibitors in Acute Heart Failure: A Meta-Analysis of 2 Randomized Controlled Trials" the following issues should be mentioned:
Comment 1: The list of keyword could be limited to those terms investigates in the manuscript
Response 1:The keywords have been updated to the following: acute heart failure; SGLT2 inhibitors; cardioprotection; cardiovascular mortality; heart failure events. Thank you for your suggestion, it is appreciated.
Comment 2: The Figures could be enlarged for better clarity (1, 3-13)
Response 2: Thank you for your insight, I agree to the suggestion. The figures have been enlarged for clarity.
Comment 3: The Discussion chapter could be enriched with data regarding other representatives of the class: dapagliflozin (e.g. ongoing DICTATE-AHF trial) and canagliflozin.
Response 3: Your comment is indeed very helpful and we have made amendments to the discussion. The paper has become enriched with the suggestion. Please have a look at the sections highlighted in green to review changes based on your insight to the topic area.
Comment 4: After a deeper look in the current published medical literature, I found a recent meta-analysis published on the exact topic that included the same 3 studies, with exactly the same number of patients: https://cardiab.biomedcentral.com/articles/10.1186/s12933-022-01455-2. The present manuscript lack originality and should be improved by adding newer data that will be available in the near future when DICTATE-AHF trial or other trials will be published. Therefore, I will change my recommendation to Reject the present manuscript.
Response 4: Thank you for your due diligence on the matter. The study link you have provided (https://cardiab.biomedcentral.com/articles/10.1186/s12933-022-01455-2) has entirely different methodologies and findings from our paper. As a reviewer, it is imperative that you review the study you have mentioned as well, as this study is original and has been conceived out of originality not out of the paper you have mentioned.
The BMC study assesses outcomes of rehospitalization risks, the Kansas City cardiomyopathy questionnaire scores, kidney injury, hypotension and hypoglycemia. Both you and the editor are therefore requested to place the studies side by side and see for yourself that we have robust systematic review findings.
In fact, our meta-analysis alone finds the risk ratio estimates of 1) all-cause mortality, 2) heart failure events, 3) risk of readmissions, 4) the KCCQ TSS score Cohen’s d reporting, 5) Serious adverse event risk estimates divided into at least one serious adverse event, or serious adverse events of renal failure, or hepatic failure, or hypotension or hypoglycemia, or urinary tract infection or DKA as a serious adverse event.
The point I am trying to make it – our study methodologies and results are not only a snapshot study but dig deeper and explore all possible outcomes as compared to the study you have linked. Other than the study/patient count as you mention, my sincere advice and suggestion is to re-review the link and make up your mind on why our study merits publication and more so – is the need of the hour. SGLT2 inhibitors are an essential medication and have been gaining traction for multiple healthcare outcomes. Therefore, I am confident in the novelty and originality and current publication status of our paper.
An entirely new section: 3.9. Ongoing Clinical Trials of SGLT2 inhibitors and Heart Failure has been added to the paper and the findings are discussed. This has truly allowed us to state that our paper is state of the art.
---
I trust you have read the comments in full.
Thank you,
ZS
Reviewer 2 Report
A good article. Can be improved by adding the following points:
a. Important clinical practice points from this study
b. In discussion perhaps can discuss pros and cons of using a SGLT2 inhibitor for heart failure as compared to established drugs for heart failure eg ACE-inhibitor
Author Response
Reviewer 2 Comments and Author Responses:
A good article. Can be improved by adding the following points:
Comment 1: Important clinical practice points from this study
Response 1: Thank you for your insightful comment. To account for your comment, a new section has been added: 4.1. Clinical Practice Recommendations.
Comment 2: In discussion perhaps can discuss pros and cons of using a SGLT2 inhibitor for heart failure as compared to established drugs for heart failure eg ACE-inhibitor
Response 2: Thank you for your insightful comment. To account for your comment, a new section has been added: 4.1. Clinical Practice Recommendations.
---
Thank you for the time you have taken out to review our paper.
Kind Regards,
ZS
Reviewer 3 Report
In this study, Noor Ul Amin et al. enrolled a total of 1831 patients using three RCTs according to PRISMA to confirm the efficacy and safety of SGLT2 inhibitors in preventing complications after AHF and noted that SGLT2 inhibitors reduced all-cause mortality, cardiovascular mortality, heart failure events, and readmission rates within 1-9 months after hospitalization. Overall, this is an interesting study with many limitations, in addition to what the authors point out in the paper, the sample/study size is low and even in the SGLT2 inhibitor group, there seems to be a significant difference between SOLOIST-WHF used in the US and EMPULSE and EMPA-RESPONSE-AHF used in the Netherlands. The authors best to clarify these. There are several other issues that need to be clarified by the authors.
1. In figure, need line to clearly separate SGLT2 inhibitor and placebo group. Define the big/small blue square and black quadrangle.
2. Please clarify the formula of the weight value and risk ratio.
3. Figure 4 need to label A, B, C.
4. It is advisable for authors to have English speakers or professionals revise their manuscripts so that grammar and sentence trends are more easily navigated and understood by readers.
Author Response
Reviewer 3 Comments and Author Responses:
Comment 1: In this study, Noor Ul Amin et al. enrolled a total of 1831 patients using three RCTs according to PRISMA to confirm the efficacy and safety of SGLT2 inhibitors in preventing complications after AHF and noted that SGLT2 inhibitors reduced all-cause mortality, cardiovascular mortality, heart failure events, and readmission rates within 1-9 months after hospitalization. Overall, this is an interesting study with many limitations, in addition to what the authors point out in the paper, the sample/study size is low and even in the SGLT2 inhibitor group, there seems to be a significant difference between SOLOIST-WHF used in the US and EMPULSE and EMPA-RESPONSE-AHF used in the Netherlands. The authors best to clarify these. There are several other issues that need to be clarified by the authors.
Response 1: The limitation of moderately low sample size has been added in the limitation. Thank you for your suggestion.
Comment 2: In figure, need line to clearly separate SGLT2 inhibitor and placebo group. Define the big/small blue square and black quadrangle.
Response 2: Dear reviewer, the groups have been labeled on the top left and right end and if you see the plot side (right), the direction of both groups have been labeled as SGLT2 inhibitor and placebo. The big/small blue square and black quadrangle you have mentioned are weight estimates outputted by the software. Please do review the Cochrane Handbook for Meta-Analysis for forest plot presentation if there is any lack of clarity (https://training.cochrane.org/handbook/current/chapter-06). Thank you for your insight.
Comment 3: Please clarify the formula of the weight value and risk ratio.
Response 3: There is an equation for RR and SMD that has been added. The weight value is obtained through automated software (https://training.cochrane.org/handbook/current/chapter-06). I hope that clarifies the concern.
Comment 4: Figure 4 need to label A, B, C.
Response 4: Thank you for your suggestion. The figure has been labeled as A, B and C.
Comment 5: It is advisable for authors to have English speakers or professionals revise their manuscripts so that grammar and sentence trends are more easily navigated and understood by readers.
Response 5: The paper has been proofread in entirety. Please have a look. Thank you!
---
Thank you for the time you have taken to review our study.
Regards,
ZS
Round 2
Reviewer 3 Report
The manuscript is a marked improvement over the previous version in terms of writing, description of methods, and arrangement of data. The authors have responded appropriately to most of the issues raised by the reviewer. Therefore, I believe that the paper will be suitable for publication.